# An Optimization Approach Considering User Utility for the PV-Storage Charging Station Planning Process

**Yingxin Liu [1,\*], Houqi Dong [1], Shengyan Wang [1], Mengxin Lan [1], Ming Zeng [1], Shuo Zhang [1], Meng Yang [2] and Shuo Yin [2]**

[1] School of Economics and Management, North China Electric Power University, Beijing 102206, China; dong_ncepu@163.com (H.D.); wangshengyan222@163.com (S.W.); lanmengxin163@163.com (M.L.); zengmingbj@vip.sina.com (M.Z.); zhangshuo@ncepu.edu.cn (S.Z.)

[2] Economic Research Institute of State Grid Henan Electric Power Company, Zhengzhou 450052, China; yangmeng0372@163.com (M.Y.); yinshuo1111@163.com (S.Y.)

[\*] Correspondence: ttkllyx1993@163.com; Tel.: +86-010-6177-3141

**Abstract:** Based on the comprehensive utilization of energy storage, photovoltaic power generation, and intelligent charging piles, photovoltaic (PV)-storage charging stations can provide green energy for electric vehicles (EVs), which can significantly improve the green level of the transportation industry. However, there are many challenges in the PV-storage charging station planning process, making it theoretically and practically significant to study approaches to planning. This paper promotes a bi-level optimization planning approach for PV-storage charging stations. First, taking PV-storage charging stations and EV users as the upper- and lower-level problems, respectively, during the planning process, a bi-level optimization model for PV-storage charging stations considering user utility is established for capacity allocation and user behavior-based electricity pricing. Second, the model is converted into a single-level mixed-integer linear programming model using the piecewise linear utility function, Karush–Kuhn–Tucker (KKT) conditions, and linearization methods. Finally, to verify the validity of the proposed model and the solution algorithm, a commercial solver is used to solve the optimization model and obtain the planning scheme. The results show that the proposed bi-level optimization model can provide a more economical and reasonable planning scheme than the single-level model, and can reduce the investment cost by 8.84%, operation and maintenance cost by 13.23%, and increase net revenue by 5.11%.

**Keywords:** PV-storage charging stations; green energy; planning process; bi-level optimization; user utility

## 1. Introduction

Advances in energy storage technology and grid intelligence and increased electric vehicle (EV) ownership have greatly promoted the development of EV charging infrastructure. However, the existing charging stations are neither low-carbon nor friendly to the distribution system because they have no energy storage facilities and must obtain electric power from the distribution network [1]. In order to take advantage of the bi-directional flow between renewable energy and energy storage systems, green photovoltaic (PV)-storage charging stations, installed with both a photovoltaic power generation system and an energy storage system, were developed based on the existing traditional charging stations. A PV-storage charging station is a microgrid that integrates the technologies of photovoltaic power generation, energy storage (ES), and smart charging station (SCS). The associated operation between photovoltaic power generation and EV charging and discharging can help promote the efficient consumption of renewable energy on-site and fulfill the EV load demand. Also, the introduction

of an energy storage system can effectively alleviate the impact of EV charging on the regional distribution network. This microgrid is an organically integrated source–storage–load system and meets the requirements for new-generation power systems: clean and efficient, green and low carbon, safe and controllable. At present, many countries, such as the United States, the United Kingdom, the Netherlands, and Malaysia, have built large-scale solar-powered EV charging stations. China has also launched some PV-storage charging EV station demonstration projects in cities such as Dongguan, Shanghai, and Qingdao. PV-storage charging stations will develop rapidly with advanced PV-storage technology and decreased economic costs in the coming days.

There are currently many studies on EV charging station planning. One study [2] summarized the main theoretical methods and research directions. Traditional charging station planning mainly aims to meet the increasing EV load demand and focuses on the siting and sizing of charging stations in the distribution network. However, with the integration of renewable energy and energy storage systems, capacity allocation and operational optimization for PV-storage charging stations have become hot research topics.

The literature on PV-storage charging station planning is mainly divided into two categories. The first category focuses on exploring the location and capacity allocation of charging stations from the perspective of distribution networks [3–5] or transportation networks [6–8], in the case of already known network structure. Charging station planning in a distribution network considers factors such as the environment, power quality [3], distribution feeder layout and availability [4], and operation safety and cost optimization [5]. When it comes to the transportation network, charging station planning considers the temporal and spatial dynamics of EV movement [6] and the spatial distribution of EVs [7], and a planning model was established using queuing theory and graph theory [8]. Considering the dual factors of distribution and traffic networks, some studies [9–12] carried out charging station planning with the objective of maximizing benefits and minimizing energy loss, while other studies [13,14] evaluated the planning results.

The second category focuses on the optimization of the internal design and energy capacity of PV-storage charging stations. This kind of study focuses mainly on internal charging stations, and rarely consider the constraints of the external network, which is usually regarded as an infinite source. Internal optimization is aimed at determining the compactness of internal facilities at charging stations, including the number of facilities and photovoltaic units and the ES capacity [15]. Commonly targeted at minimizing operation cost, the optimization model was established based on constraints such as operation, cost, and equipment utilization [16] and then was solved using corresponding particle swarm optimization algorithms [17] and NSGA-II (Non-dominated Sorting Genetic II Algorithm) [18]. In addition to cost, another study [19] also considered queuing time. It can be easily seen from the above studies that current PV-storage charging station planning rarely considers uncertain factors such as distributed generation, user behavior, and electricity price.

This paper, therefore, aims to study the internal optimization of PV-storage charging stations under uncertain conditions. Taking the uncertain factors—the charging station operator (CSO) and EV users—as the upper- and lower-level problems, a user behavior-based bi-level optimization model for the PV-storage model was established to determine the capacity allocation and electricity pricing. In the upper-level model, EV charging capacity constraints are considered, and there is an assumption that each EV user charges at a charging station only during a single time period t, meaning the EV can be charged to the maximum storage capacity during that time period, in order to simplify the planning problem. In the lower-level model, the real-time electricity price is considered to calculate the expected revenue of PV-storage charging stations. The model was then converted into a single-level mixed-integer linear programming using the piecewise linear utility function, Karush–Kuhn–Tucker (KKT) conditions, and linearization methods. The obtained linear programming problem was solved to compare and analyze the quantitative influence of uncertain variables on charging station planning.

## 2. Bi-Level Optimization Model for PV-Storage Charging Station Planning

### 2.1. PV-Storage Charging Station System

The PV-storage charging station system, the upper-level problem studied in this paper, is shown in Figure 1. A PV-storage charging station is usually composed of multiple EV charging facilities (CFs), energy storage (ES) units, and photovoltaic units. A charging station that is connected to an external distribution network via a power electronic converter can obtain its energy supply from the main power grid in case of insufficient power generation. ES units are installed in the system in order to make full use of renewable energy (RES). When the PV output power is higher than the charging demand of the connected EV, the ES can start power charging to store excess energy. When the PV output is insufficient, the ES can perform power discharging to supply energy. Therefore, the PV-storage charging station can achieve minimal energy consumption by coordinately controlling the ES and PV output. All of the above components are assumed to be integrated into the charging station based on the AC interface and controlled centrally by the system operator.

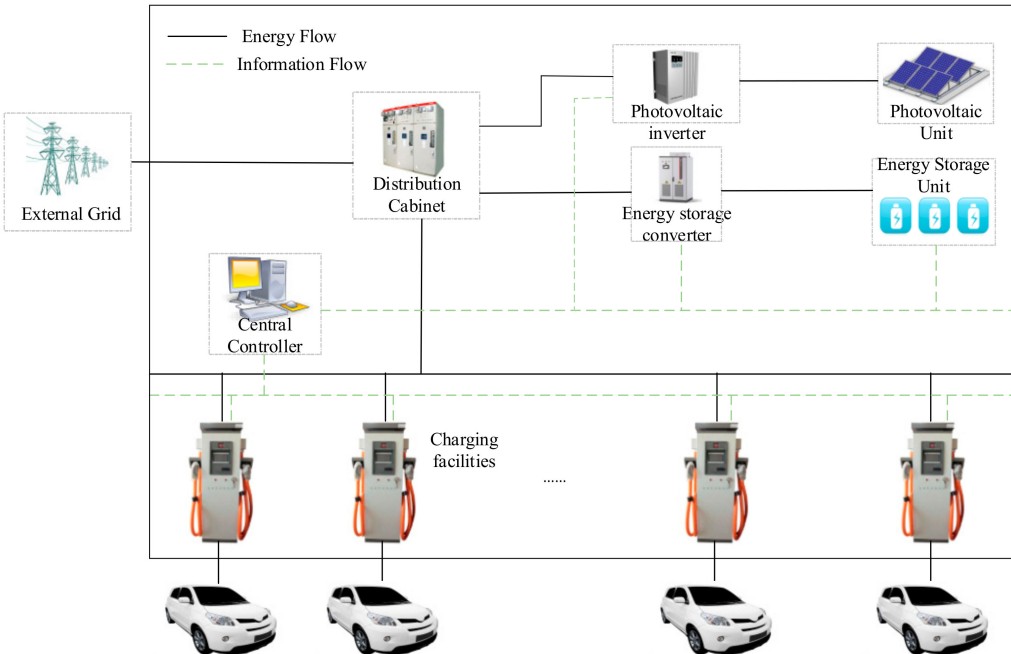

**Figure 1.** Internal structure of photovoltaic (PV)-storage charging station and energy and information flows.

Both the charging station operator (CSO) and EV users are considered in this paper. The CSO is responsible for the construction and operation of the charging station and will consider the EV users' selection strategy in charging storage planning. The CSO considers the construction cost, operation, and maintenance cost, and power sales revenue to determine capacity allocation and electricity pricing, based on the premise of satisfying the internal balance and security constraints. EV users can choose from the electricity pricing schemes offered by the CSO to obtain better economic benefits.

This paper assumes that the investment and operation of the PV-storage charging station are performed by the CSO. The PV-storage charging station earns revenue by providing EV users with energy from PV units or the external power grid. In addition, the CSO can also profit from trading with the power grid via PV power generation in a deregulated environment. The electricity exchange price is determined through bilateral negotiation between the CSO and the main power grid and remains unchanged during the contract period. The operational performance of the charging station varies with floating EV charge demand. The CSO must fully consider the potential response of EV users,

participate in the design of electricity pricing, and coordinately determine the optimal configuration and control strategies in order to maximize profits.

On the other hand, EV users have to decide on their charging load at the charging station each time they charge (provided that the minimum charging demand is large enough to power the next driving distance), and their charging demand is affected by the electricity price. Therefore, the optimal decision for EV users depends not only on their own preference but also on the CSO's electricity pricing, which may conflict with their goals.

The optimal planning and decision problem for the CSO can be expressed as a bi-level planning model, as shown in Figure 2. The upper-level problem represents the CSO's decision in relation to the PV, ES, CS configuration, and electricity pricing to maximize the expected revenue. The lower-level problem considers the CSO's pricing scheme and predicts the EV users' response by calculating the optimal charging benefits for each user. The price implemented by the CSO is in real-time and can be adjusted to influence the users' behavior so that they can be encouraged to actively respond to the RES output curve. The construct of this model will help to obtain an optimal capacity configuration and price scheme.

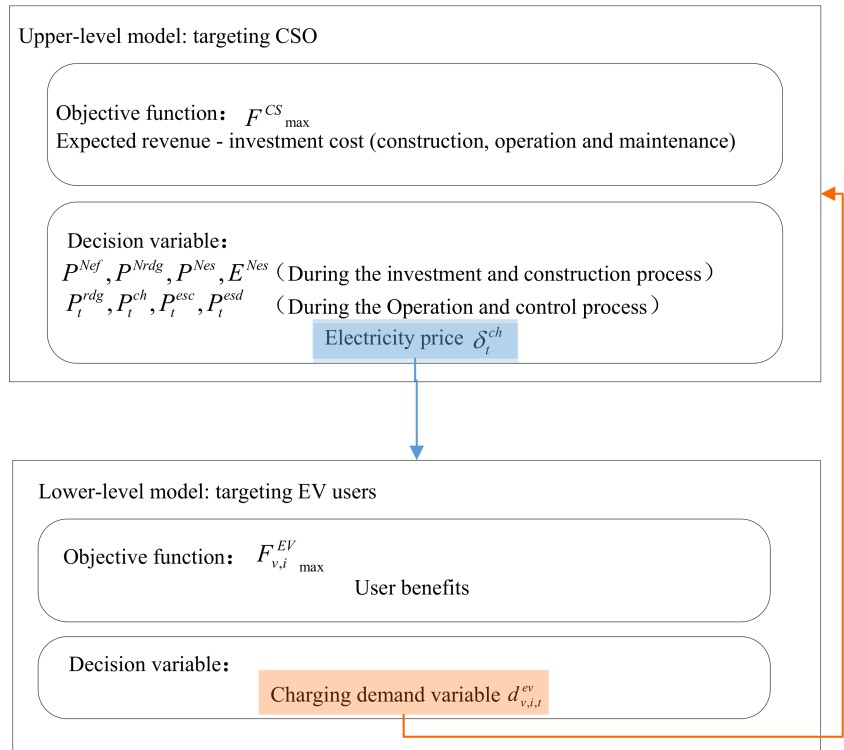

**Figure 2.** Bi-level model framework for optimizing the design and decision-making of PV-storage charging stations. CSO, charging station operator. EV, electric vehicle.

### 2.2. Upper-Level Model of PV-Storage Charging Station Planning

### 2.2.1. Objective Function

According to the description of the model framework in the previous section, the upper-level optimization problem involves optimal configuration and pricing of PV-storage charging stations, i.e., determining PV, ES, and CS capacity and pricing. The objective function for the upper-level model is shown in Equation (1):

$$\max_{\substack{P^{Nrdg}, P^{Ncf}, P^{Nes}, E^{Nes}, \delta_t^{ch} \\ P_t^{rdg}, P_t^{esc}, P_t^{esd}, P_t^{ch}, P_t^{int}}} F^{CS} = B^{Ope} - C^{Inv} \tag{1}$$

$$C^{Inv} = k^{cf} c^{cf} P^{Ncf} + k^{rdg} c^{rdg} P^{Nrdg} + k^{es} (c^{esp} P^{Nes} + c^{ese} E^{Nes}) \tag{2}$$

$$B^{Ope} = \theta \cdot \sum_{t=1}^{T} \left( \delta_t^{ch} P_t^{ch} - \delta_t^{int} P_t^{int} \right) \Delta t - (c^{cfm} P^{Ncf} + c^{rdgm} P^{Nrdg} + c^{esm} E^{Nes}) \tag{3}$$

$$k = \zeta (1 + \zeta)^d / \left[ (1 + \zeta)^d - 1 \right] \tag{4}$$

In the bi-level model, the objective function for the upper-level model is the maximum revenue of the CSO, and expected revenue is the difference between the expected revenue ($B^{Ope}$) and the investment cost ($C^{Inv}$). In Equation (2), $C^{Inv}$ represents the investment cost, which is correlated with the construction cost of the CF, PV, ES facilities, land lease, and other relevant expenses. $P^{Ncf}$, $P^{Nrdg}$, $P^{Nes}$, and $E^{Nes}$ represent the rated installed power of CF and PV, ES, and the installed storage capacity of ES, respectively. $c^{cf}$, $c^{rdg}$, $c^{esp}$, and $c^{ese}$ represent the unit investment cost of CF, PV, and ES facilities. $k^{cf}$, $k^{rdg}$ and $k^{es}$ represent the capital recovery factor of CF, PV, and ES, which can be calculated by Equation (4). In Equation (3), $B^{Ope}$ represents expected revenue, including the charging service revenue from EV users and revenue from interactive transactions with the grid. $\theta$ represents the number of days in a year. $\delta_t^{ch}$ and $\delta_t^{int}$ represent the electricity price per kWh sold by the CSO and traded with the main grid, respectively. $P_t^{ch}$ and $P_t^{int}$ represent the charging power of EV users at time $t$ and power during the interactive transaction with the main power grid. $\Delta t$ represents the time of each period, which is defined 0.5 h in this paper. When $P_t^{int}$ is positive, electricity is purchased from the main power grid. When $P_t^{int}$ is negative, electricity is sold to the main power grid. $c^{cfm}$, $c^{rdgm}$ and $c^{esm}$ represent the annual unit operation cost of the CF, PV, and ES facilities, respectively. In Equation (4), k represents the capital recovery factor, and d and $\zeta$ represent the facility lifetime and discount rate, respectively.

### 2.2.2. Constraints

The CSO needs to consider constraints during facility construction and operation, including constraints of investment, balance, and security. The specific constraints are shown in Equations (5)–(19). Equations (5)–(9) are investment and price constraints:

$$0 \leq P^{Ncf} \leq P^{Ncf}_{max} \tag{5}$$

$$0 \leq P^{Nrdg} \leq P^{Nrdg}_{max} \tag{6}$$

$$0 \leq P^{Nes} \leq P^{Nes}_{max} \tag{7}$$

$$0 \leq E^{Nes} \leq E^{Nes}_{max} \tag{8}$$

$$0 \leq \delta_t^{ch} \leq \delta_{max}^{ch} \ \forall t = 1, \cdots, T \tag{9}$$

Equations (5)–(7) represent the maximum installable rated power constraints for CF, PV, and ES in the charging station, and their investment capacity has an upper limit due to capital or space limitations. Equation (8) represents the maximum installable storage capacity of ES. Equation (9) represents the price constraints by the CSO that can help ensure the stability of the EV charging market. Equations (10)–(19) are operation constraints:

$$-P_{max}^{tr} \leq P_t^{int} \leq P_{max}^{tr} \ \forall t = 1, \cdots, T \tag{10}$$

$$0 \leq P_t^{rdg} \leq P^{Nrdg} \gamma_t^{rdg} \ \forall t = 1, \cdots, T \tag{11}$$

$$0 \leq P_t^{ch} \leq P^{Ncf} \ \forall t = 1, \cdots, T \tag{12}$$

$$0 \leq P_t^{esc} \leq P^{Nes} \ \forall t = 1, \cdots, T \tag{13}$$

$$0 \leq P_t^{esd} \leq P^{Nes} \ \forall t = 1, \cdots, T \tag{14}$$

Equation (10) specifies that the exchange power between the charging station and the main power grid should not exceed the substation transformer capacity. Equations (11) and (12) specify the maximum PV scheduling power and CF capacity constraint, where $\gamma_t^{rdg}$ represents the photovoltaic output factor. Equations (13) and (14) represent the charging and discharging constraints of the ES, where $P_t^{esc}$ and $P_t^{esd}$ represent charging and discharging power, respectively, at time $t$.

$$E_t^{es} = E_{t-1}^{es} + P_t^{esc}\eta^{esc}\Delta t - P_t^{esd}\Delta t/\eta^{esd} \quad \forall t = 1, \cdots, T \tag{15}$$

$$E^{Nes}SOC_{min}^{es} \leq E_t^{es} \leq E^{Nes}SOC_{max}^{es} \quad \forall t = 1, \cdots, T \tag{16}$$

$$E_0^{es} = E_T^{es} \tag{17}$$

$$P_t^{rdg} + P_t^{int} = P_t^{ch} + P_t^{esc} - P_t^{esd} \quad \forall t = 1, \cdots, T \tag{18}$$

$$P_t^{ch}\eta^{cf}\Delta t = \sum_{v \in \Omega_V} d_{v,t}^{ev} f_{v,t} \quad \forall t = 1, \cdots, T \tag{19}$$

Equations (15) and (16) show the energy change characteristics and their charge and discharge state (SOC) constraints during operation of the ES unit, where $SOC_{min}^{es}$ and $SOC_{max}^{es}$ represent the minimum and maximum charge and discharge states, and $\eta^{esc}$ and $\eta^{esd}$ represent charge and discharge efficiency. Equation (17) specifies that the available capacity ($E_T^{es}$) of the ES at the end of the dispatching operation must be consistent with the start capacity ($E_0^{es}$) [20] to ensure the sustainable performance of the charging station. Equation (18) is the power balance constraint of the PV-storage charging station. Equation (19) shows that the power load provided by the CSO must always satisfy the charging demand of the EV user, where $d_{v,t}^{ev}$ represents the power demand of a v-type electric vehicle at time $t$, and $f_{v,t}$ represents the number of v-type vehicles in use. Equation (19) is a constraint that connects the upper and lower levels, including the upper variable $P_t^{ch}$ and lower decision variable, where $\Delta t$ represents the time scale, set as 0.5 h in this paper. For the sake of simplicity, it is assumed that each EV user charges in the charging station only during a single time period t, regardless of the charging behavior across multiple time periods.

*2.3. Lower-Level Model for PV-Storage Charging Station Planning*

2.3.1. Objective Function

The lower-level optimization problem in Figure 2 is the behavior selection problem of EV users based on real-time pricing. The charging demand at each period is determined. The objective function of the lower-level problem is shown in Equation (20):

$$\underset{d_{v,t}^{ev}}{Max}F_v^{EV} = [U_v(d_{v,t}^{ev}) - \delta_t^{ch}d_{v,t}^{ev}] \tag{20}$$

The objective function for the lower-level problem is the maximum EV user benefits, and the benefits are expressed as the difference between user utility $U_v(d_{v,t}^{ev})$ and the electricity charging cost. $U_v$ in the equation represents the utility function of the electric vehicle user, and its specific form will be described in detail in the next section, and the other symbols have the same meanings as in the previous section.

2.3.2. Constraints

Equation (21) is the EV power demand constraint, which is used to ensure that the user's charging demand meets the minimum charging load and maximum charging limit, i.e., the EV capacity is kept between the minimum mileage power and the rated battery capacity.

$$d_{v,min}^{ev} \leq d_{v,t}^{ev} \leq d_{v,max}^{ev} \tag{21}$$

The essence of the lower-level optimization is that EV users make their choices according to the electricity price, i.e., users just need to consider their own benefits and costs and do not need to consider the constraints of the charging station and the power grid, since the capacity configuration and price-based optimal charging load in each period are determined by the CSO in the upper-level model.

## 3. Algorithms for Bi-Level PV-Storage Charging Station Planning

### 3.1. KKT Algorithm Analysis

In the previous section, a bi-level planning model was established to solve user behavior-based PV-storage charging station planning, but the specific form of the benefits function for the lower-level problem was not described. This section elaborates EV user benefits using the piecewise linear function, where the lower-level problem is linearized and, based on this, the KKT conditions are used to convert the bi-level optimization problem into a single-level optimization problem. Then linear approximation is used to transform the single-level problem into a mixed-integer linear programming problem that is finally solved using the classical solutions.

As shown in Figure 3, the bi-level optimization problem is converted into a single-level mixed-integer linear programming problem through the following three steps: first, the lower-level function is piecewise linearized to represent user utility. Second, the KKT conditions, which are used to replace the lower-level problem, are included in the upper-level problem, thus obtaining an equivalent single-level problem. Finally, the nonlinear terms in the obtained single-level problem are linearized to obtain a single-level mixed-integer linear program. In this way, the original problem is transformed and can be efficiently solved.

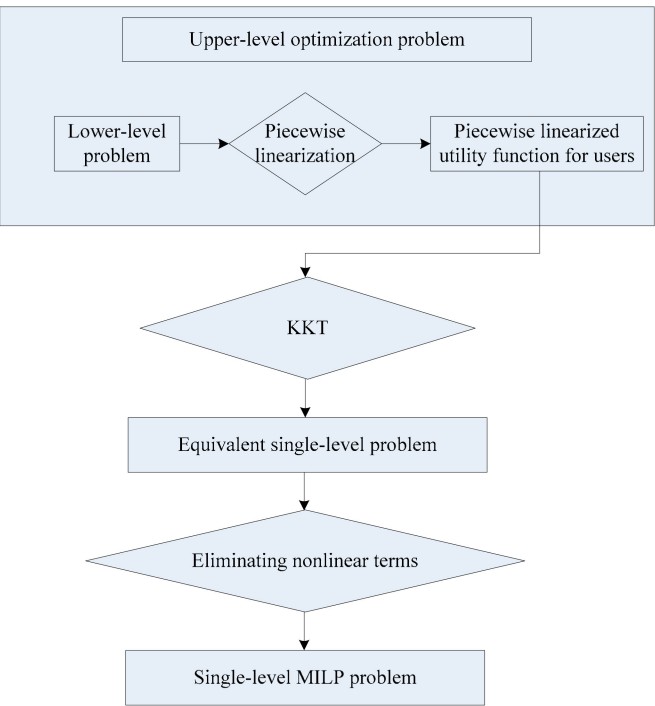

**Figure 3.** Bi-level planning algorithm flowchart. KKT, Karush–Kuhn–Tucker. MILP, mixed-integer linear programming.

### 3.2. Bi-Level Optimization Model Solution

#### 3.2.1. Linear Description of Lower-Level Problem

The utility function is used to measure consumer satisfaction as a function of consumption of a vested commodity combination. The consumers in this paper are EV users, and the commodity is

charging power. According to the utility function, the utility increases as a function of increasing purchased commodities, while the marginal utility of unit commodity decreases.

As shown in Figure 4, this paper assumes that EV user utility is a piecewise linear function of charge capacity, given that the piecewise function is a typical function representing the relationship between welfare utility and energy consumption [21,22], and linearization of the EV user benefits utility can be applied to the proposed bi-level optimization model and solution method. Without loss of generality, this paper assumes that the segmentation function is divided into five segments (M = 5), and each segment corresponds to a predetermined marginal utility value. $d^{ev}_{v,t,m}$ represents the charging demand of EV users in each segment, and the total utility from the CSO can be expressed as piecewise linearized using Equations (22) and (23):

$$\sum_{m=1}^{M} d^{ev}_{v,t,m} = d^{ev}_{v,t} \tag{22}$$

$$U_v(d^{ev}_{v,t}) = \sum_{m=1}^{M} u^{ev}_{v,m} d^{ev}_{v,t,m} \tag{23}$$

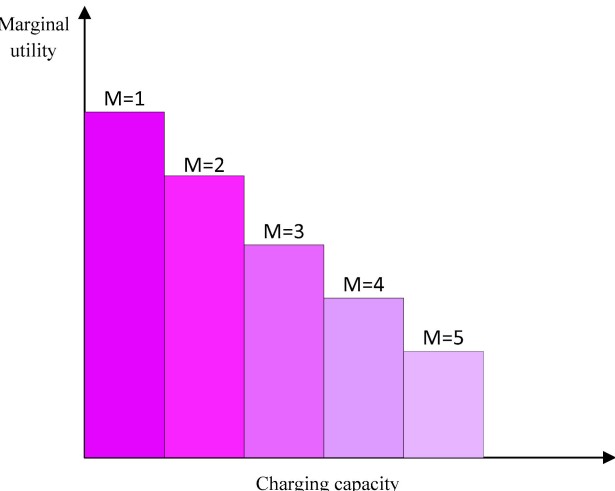

**Figure 4.** EV user utility function graph.

The original lower-level model can be expressed as Equation (24):

$$\underset{d^{ev}_{v,t,m}, d^{ev}_{v,t}}{\text{Maximize}} F^{EV}_v = \delta^{ch}_t d^{ev}_{v,t} - \sum_{m=1}^{M} \left( u^{ev}_{v,m} d^{ev}_{v,t,m} \right) \tag{24}$$

Equations (25)–(29) are charging constraints for EV users. Equation (25) is a logical constraint that associates the decision variable $d^{ev}_{v,t}$ with its piecewise form $d^{ev}_{v,t,m}$. The total capacity of EV users is assumed to meet the daily driving distance requirements without exceeding the battery capacity.

$$\sum_{m=1}^{M} d^{ev}_{v,t,m} = d^{ev}_{v,t} : \lambda_{v,t} \tag{25}$$

$$d^{ev}_{v,t} - d^{ev}_{v,max} \leq 0 : \mu^1_{v,t} \tag{26}$$

$$-d^{ev}_{v,t} + d^{ev}_{v,\min} \leq 0 : \mu^2_{v,t} \tag{27}$$

$$d^{ev}_{v,t,m} - d^{ub}_{v,t,m} \leq 0 \ \forall m \in M : \mu^3_{v,t,m} \tag{28}$$

$$d_{v,t,m}^{ev} \geq 0 \ \forall m \in M : \mu_{v,t,m}^{4} \tag{29}$$

$$d_{v,max}^{ev} = E_{rated}^{ev}\left(SOC_{max}^{ev} - SOC_{v,in}^{ev}\right) \tag{30}$$

$$d_{v,min}^{ev} = l_{v}^{tot}\varepsilon^{ev} + E_{rated}^{ev}\left(SOC_{min}^{ev} - SOC_{v,in}^{ev}\right) \tag{31}$$

where $E_{rated}^{ev}$ and $\varepsilon^{ev}$ represent EV battery capacity and capacity consumption per kilometer, $l_{v}^{tot}$ represents the expected the daily travel distance of v-type EVs, and $SOC_{min}^{ev}$, $SOC_{max}^{ev}$ and $SOC_{v,in}^{ev}$ represent the minimum and maximum SOCs of EV batteries and the initial SOC at the charging station. Equation (28) indicates that the EVs' charge capacity in each segment is smaller than the upper limit of this segment. Equation (29) indicates that the charge capacity of each segment is positive. It should be noted that the subsequent variable of each equation is its dual variable.

Based on this, the linear expression of the bi-level problem can be obtained.

### 3.2.2. Reducing a Bi-Level Problem to a Simple Level Problem

The lower-level problem in the previous section was linearized. In the proposed bi-level problem, according to its structural and form transformation, the upper-level variable can be considered as a parameter in the lower-level problem, and for a given variable $\delta_t$, each lower-level problem is a linear optimization problem that is continuous and convex in structure. Therefore, the KKT conditions can be used to replace the lower-level problems in Equations (24)–(29) with their corresponding KKT optimality conditions and then be included in the upper-level problem. By doing this, the original bi-level model is transformed into an equivalent single-level problem [23].

The equations for the KKT conditions are as follows:

Equation (32) represents the Lagrangian function of the lower-level problem, and $\lambda_{v,t}$, $\mu_{t}^{1}$, $\mu_{t}^{2}$, $\mu_{t,m}^{3}$, and $\mu_{t,m}^{4}$ represent the Lagrange multipliers of the corresponding lower-level constraints. Equations (35)–(38) are relaxed complementary constraints, where $0 \leq \mu \perp d \geq 0$ represents $0 \leq \mu, d \geq 0, \mu d = 0$. It should be noted that Equations (35)–(38) have a bilinear term by multiplying the dual variable and the original variable.

$$L(d_{v,t}^{ev}, d_{v,i,t,m}^{ev}, \lambda, \mu) = d_{v,t}^{ev}\delta_{t}^{ch} - \sum_{m=1}^{M} d_{v,t,m}^{ev}u_{v,t,m}^{ev} + \lambda_{v,t}\left(d_{v,t}^{ev} - \sum_{m=1}^{M} d_{v,t,m}^{ev}\right)$$
$$+\mu_{v,t}^{ev}\left(d_{v,t}^{ev} - d_{v,t,min}^{ev}\right) + \mu_{v,t}^{2}\left(-d_{v,t}^{ev} + d_{v,t,max}^{ev}\right) + \mu_{v,t,m}^{3}\left(d_{v,t,m}^{ev} - d_{v,t,m}^{max}\right) - \mu_{v,t,m}^{4}d_{v,t,m}^{ev} \tag{32}$$

$$\frac{\delta L}{\delta d_{v,t}^{ev}} = \delta_{t}^{ch} + \lambda_{v,t} + \mu_{v,t}^{1} - \mu_{v,t}^{2} = 0 \ \forall t \in T \tag{33}$$

$$\frac{\delta L}{\delta d_{v,t,m}^{ev}} = u_{v,t,m}^{ev} + \lambda_{v,t} + \mu_{v,t,m}^{3} - \mu_{v,t,m}^{4} = 0 \ \forall t \in T, \ m \in M \tag{34}$$

$$0 \leq \mu_{v,t}^{1} \perp (d_{v,t}^{ev} - d_{v,t,min}^{ev}) \geq 0 \ \forall t \in T \tag{35}$$

$$0 \leq \mu_{v,t}^{2} \perp (-d_{v,t}^{ev} + d_{v,t,max}^{ev}) \geq 0 \ \forall t \in T \tag{36}$$

$$0 \leq \mu_{v,t,m}^{3} \perp (d_{v,t,m}^{ev} - d_{v,t,m}^{ub}) \geq 0 \ \forall t \in T, \ m \in M \tag{37}$$

$$0 \leq \mu_{v,t,m}^{4} \perp d_{v,t,m}^{ev} \geq 0 \ \forall t \in T, \ m \in M \tag{38}$$

By substituting Equations (24)–(29) for the lower-level problem with Equations (33)–(38), a single-level model equivalent to the bi-level optimization can be obtained.

### 3.2.3. Linear Description of Lower-Level Problem

We can transform the bi-level model into a single-level problem, but this single-level problem is bilinear and has nonlinear terms, and cannot obtain an exact solution using the classical algorithm. Therefore, the nonlinear terms in the single-level problem described in the previous section should be

linearized. The nonlinear terms in the single-level constraints are generated when the KKT conditions are introduced to achieve single-level transformation. The relaxing complementarity conditions introduced in the constraint have nonlinear terms by multiplying two variables, and these nonlinear terms can be transformed into a mixed-integer linear program [24]. The specific transformation for $0 \leq \mu \perp d \geq 0$ is as follows:

$$
\begin{aligned}
\mu &\geq 0, d \geq 0 \\
\mu &\leq (1-w)M \\
d &\leq wM \\
w &\in \{0, 1\}
\end{aligned}
\tag{39}
$$

where $M$ is a sufficiently large constant, and $w$ is a variable between 0 and 1. It should be pointed out that since the value of $M$ will affect the accuracy of the problem and the efficiency of the calculation, decision-makers need to make appropriate choices. Some studies have given alternatives.

By using the above method, the nonlinear complementarity constraints in Equations (35)–(38) can be rewritten as follows:

$$
\mu_{v,t}^1 \geq 0, d_{v,t,\min}^{ev} - d_{v,t}^{ev} \geq 0
\tag{40}
$$

$$
\mu_{v,t}^1 \leq \left(1 - w_{v,t}^1\right)M_1, d_{v,t,\min}^{ev} - d_{v,t}^{ev} \leq w_{v,t}^1 M_1
\tag{41}
$$

$$
\mu_{v,t}^2 \geq 0, d_{v,t}^{ev} - d_{v,t,\max}^{ev} \geq 0
\tag{42}
$$

$$
\mu_{v,t}^2 \leq \left(1 - w_{v,t}^2\right)M_2, d_{v,t}^{ev} - d_{v,t,\max}^{ev} \leq w_{v,t}^2 M_2
\tag{43}
$$

$$
\mu_{v,t,m}^3 \geq 0, d_{v,t,m}^{ub} - d_{v,t,m}^{ev} \geq 0
\tag{44}
$$

$$
\mu_{v,t}^3 \leq \left(1 - w_{v,t}^3\right)M_3, d_{v,t,m}^{ub} - d_{v,t,m}^{ev} \leq w_{v,t,m}^3 M_3
\tag{45}
$$

$$
\mu_{v,t,m}^4 \geq 0, d_{v,t,m}^{ev} \geq 0
\tag{46}
$$

$$
\mu_{v,t}^4 \leq \left(1 - w_{v,t}^4\right)M_4, d_{v,t,m}^{ev} \leq w_{v,t,m}^4 M_4
\tag{47}
$$

$$
w_{v,t}^1, w_{v,t}^2, w_{v,t,m}^3, w_{v,t,m}^4 \in \{0,1\}
\tag{48}
$$

In addition to the upper-level objective function, the above process linearizes the nonlinear constraints using an equivalent mixed-integer linear programming model. Here we linearized the nonlinear terms in the upper-level objective function using a classical linear approximation method. Owing to its simplicity and easy access to a solution, this method has been fully applied in some studies [25,26]. The specific method is as follows.

According to the linearized model and assuming $z = a \times b$ represents a bilinear term, $a \in [a_{\min}, a_{\max}]$, $b \in [b_{\min}, b_{\max}]$, the bilinear term $z = a \times b$ can be linearized by the following form:

$$
\begin{aligned}
z &\geq a_{\min}b + b_{\min}a - a_{\min}b_{\min} \\
z &\geq a_{\max}b + b_{\max}a - a_{\max}b_{\max} \\
z &\leq a_{\min}b + b_{\max}a - a_{\min}b_{\max} \\
z &\leq a_{\max}b + b_{\min}a - a_{\max}b_{\min}
\end{aligned}
\tag{49}
$$

Applying this method to the upper-level objective function, the converted objective function can be expressed as:

$$
B^{Ope} = \theta \cdot \sum_{t=1}^{T} \left(z_t^{ch} - \delta_t^{int} P_t^{int}\right)\Delta t - \left(c^{cfm}P^{Ncf} + c^{rdgm}P^{Nrdg} + c^{esm}E^{Nes}\right)
\tag{50}
$$

where constraints are added correspondingly:

$$
z_t^{ch} \geq \delta_{\max}^{ch} P_t^{ch} + z_t^{cf} - \delta_{\max}^{ch} P^{Ncf} \quad \forall t = 1, \cdots, T
\tag{51}
$$

$$z_t^{ch} \leq z_t^{cf} \quad \forall t = 1, \cdots, T \tag{52}$$

$$z_t^{ch} \leq \delta_{max}^{ch} P_t^{ch} \quad \forall t = 1, \cdots, T \tag{53}$$

$$z_t^{cf} \geq \delta_{max}^{ch} P^{Ncf} + P_{max}^{Ncf} \delta_t^{ch} - \delta_{max}^{ch} P_{max}^{Ncf} \quad \forall t = 1, \cdots, T \tag{54}$$

$$z_t^{cf} \leq P_{max}^{Ncf} \delta_t^{ch} \quad \forall t = 1, \cdots, T \tag{55}$$

$$z_t^{cf} \leq \delta_{max}^{ch} P^{Ncf} \quad \forall t = 1, \cdots, T \tag{56}$$

$$z_t^{ch} \geq 0 \quad \forall t = 1, \cdots, T \tag{57}$$

$$z_t^{cf} \geq 0 \quad \forall t = 1, \cdots, T \tag{58}$$

The nonlinear terms in the upper-level objective function are effectively linearized using the above method. As a result, the nonlinear bi-level optimization programming model is transformed into a single-level mixed-integer linear program, which can be effectively solved through commercial solvers such as CPLEX (12.7.1, IBM, Armonk, NY, USA, 2018).

## 4. Case Analysis

Numerical simulations of the model mentioned in the first two sections were conducted to obtain planning results of power pricing and capacity allocation. The planning results of the existing bi-level optimization model were compared with those of the common single-level optimization model and their differences were summarized. The superiority of the bi-level model was verified.

### 4.1. Basic Data

In this paper, solar photovoltaic panels and lithium-ion batteries were used as photovoltaic power generation and energy storage equipment. Relevant data shown in Table 1 were obtained from the website and merchant research. Assuming that the total area of the PV-storage charging station is 3000 m², the maximum installation power and capacity of CF, PV, and ES is 5000 kW, 500 kW, and 900 kW (2500 kWh), respectively, and the basic discount rate is 6%, based on compound interest.

**Table 1.** Basic parameters for facilities. SOC, charge and discharge state.

| Equipment Type | Technical Parameters | Cost Parameters |
|---|---|---|
| Photovoltaic | d = 20 years | $c^{rdg}$ = \$870/kW $c^{rdgm}$ = \$12/kW/years |
| Lithium battery | d = 20 years $\eta^{esc} = \eta^{esd}$ = 93% $SOC_{min}^{es}$ = 30% $SOC_{max}^{es}$ = 90% | $c^{esp}$ = \$200/kW $c^{ese}$ = \$143/kWh $c^{esm}$ = \$0.8/kWh/years |
| Charging pile | d = 20 years $\eta^{cf}$ = 95% | $c^{cf}$ = \$100/kW $c^{cfm}$ = \$6/kW/years |

The transformer–capacitor connecting the PV-storage charging station to the external power grid is set as 4000 kVA, the real-time price of a certain power market is set as the real-time electricity price, and the output parameter curve of a certain PV plant is used the PV output factor. The specific parameters are shown in Figure 5.

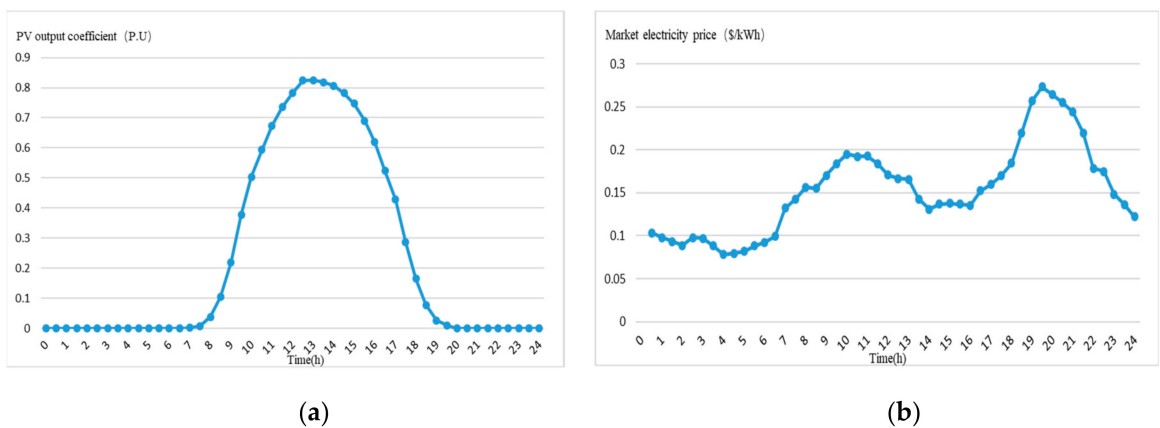

(**a**)　　　　　　　　　　　　　　　　　　　　　(**b**)

**Figure 5.** (**a**) Photovoltaic output coefficient and (**b**) market electricity price parameter.

For simplicity, the simulation example uses Nissan's electric vehicles to represent the EV user community. Each EV has a battery capacity of 40 kWh and a power consumption of 0.18 kWh/km. The lower and upper limits of SOC are set as 30% and 90%, respectively. To be consistent with the proposed model framework, EV users in the charging station system are assumed to be classified into three categories based on daily driving distance: remote range (LR), medium-range (MR), and short-range (SR). In addition, the CSO can predict EV traffic and the proportion of EV users in the three driving distance groups at the charging station at different times of the day. The numbers of LR, MR, and SR EV users are shown in Figure 6, and their parameters are shown in Table 2.

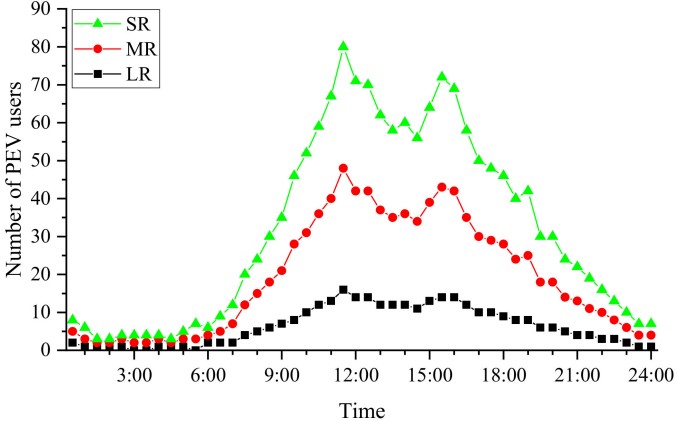

**Figure 6.** Numbers of different types of electric vehicle users.

**Table 2.** Basic parameters of different types of electric vehicle (EV) users. SR, short-range. MR, medium-range. LR, long-range.

| Parameter | User Type | | |
| --- | --- | --- | --- |
| | SR | MR | LR |
| $SOC_{in}^{ev}$ | 30% | 40% | 50% |
| $l^{tot}$ | 20 km | 50 km | 100 km |

### 4.2. Simulation Example Results

Based on the above basic data, the commercial solver CPLEX (12.7.1, IBM, Armonk, New York, USA, 2018) was used to perform the example simulation in the MATLAB environment (2018a, MathWorks, Natick, MA, USA, 2018), and the obtained PV-storage charging planning results and economic cost are shown in Table 3. The real-time electricity pricing profile is shown in Figure 7.

**Table 3.** PV-storage charging station planning results. CF, charging facility. ES, energy storage.

| | PV Installed Power (kW) | CF Installed Power (kW) | ES Installed Power and Storage Capacity (kW/kWh) | Daily Electricity Sales (kWh) |
|---|---|---|---|---|
| **Planning Results** | 500 | 862 | 616/900 | 8764 |
| | **Annual Investment Cost ($)** | **Annual Operation and Maintenance Cost ($)** | **Annual Revenue ($)** | **Annual Net Revenue ($)** |
| | 79,650 | 11,839 | 844,029 | 752,485 |

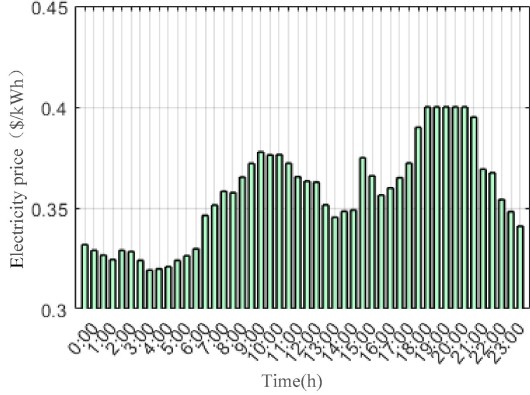

**Figure 7.** Real-time electricity price profile.

It can be seen from Table 3 that the daily sales of the PV-storage charging station is 8.764 MWh, with an annual net revenue of $752,485. The photovoltaic installation is at the boundary of the maximum installed capacity, indicating that during operation of the charging station, photovoltaic power generation can achieve higher revenue compared with purchasing electricity from the main power grid. Therefore, the planning results show that under sufficient load, the PV-storage charging station should use more available space to install photovoltaics. In this operation, the photovoltaic output accounts for nearly 30% and can be fully consumed owing to more energy storage facilities, indicating that overall advances in energy storage technology can help cost reduction.

It can be seen from Figure 7 (the real-time electricity price profile) that the highest electricity price in a day is $0.40/kWh, which occurs between 19:00 and 21:00, while the lowest electricity price is $0.32/kWh, which occurs between 03:00 and 04:00. From the perspective of the price changing trend, the electricity price peaks from 09:00 to 11:00 and from 19:00 to 21:00, during which it reaches the highest limit peak. There is a small fluctuation in the afternoon, which is consistent with the charging behavior of EV users. EV users charge their batteries mainly in the morning and evening, so the charging load is greater during those periods. The photovoltaic output is low and even reaches zero at night, and the electricity from the main power grid is also priced higher during this period. Therefore, the PV-storage charging station has to turn to the main power grid or energy storage system to meet the users' load demand. At this time, the CSO should set a higher electricity price to ensure the profitability of the system.

The planning results of the simulation of the PV-storage charging station system are shown in Figure 8. The red bars represent the power demand of EV users, the blue and orange bars represent the charge and discharge states of the ES and the charge and discharge amounts, respectively, and the purple bars represent PV output. If a green bar is positive, it means the charging station purchased electricity from the main grid. If the green bar is negative, it means the charging station sold electricity to the main grid. The figure is based on the internal power balance of PV-storage charging stations. The area above the time axis represents the power source of the charging station, while the area below represents the power output. Total power generation is equal to total power consumption, so the figure has vertical symmetry.

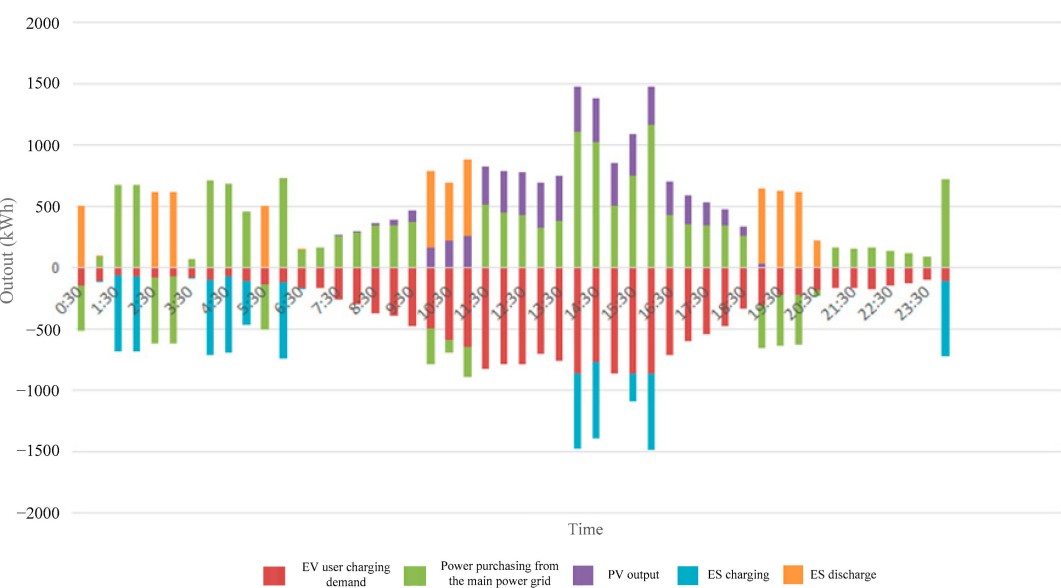

**Figure 8.** Operation strategy of PV-storage charging station.

It can be seen from Figure 8 that the EV load is greater in the morning and the evening, and smaller at night. From the perspective of time, the energy storage system works more between 22:00 and 05:00 the next day, because the electricity price of the main power grid is lower at this time, and the CSO purchases electricity from the main grid and continuously charges electricity in the ES system to maintain storage power. In addition, three ES discharges occur between 00:00 and 02:00 at night. Since the night electricity price is floating, the CSO is profitable by using low-price charging and selling high-priced electricity at the next moment. In general, the charging station performs ES at night, and between 05:00 and 09:00, the EV load is relatively small and the photovoltaic power generation is also very small. At this time, the load is mainly satisfied by the power supply purchased from the main power grid without using the stored energy in the ES system. However, during the peak charging period in the morning, the CSO calls up a large amount of nighttime energy storage and uses PV power to meet the EV charging demand, while the charging station can transfer electricity to the external grid until the SOC of the ES reaches the lower limit. This is the main period when the charging station can generate a profit. From 11:00 to 13:00, the CSO directly purchases electricity from the main power grid and photovoltaic power to meet the EV user load without activating the ES system. The ES is charged to store energy until 13:00–15:00 when the PV output is large and the electricity purchased from the main power grid is low-priced. The situation from 16:00 to 18:00 is similar to that from 11:00 to 13:00. From 18:00 to 21:00, the CSO meets the needs of EV users by discharging the energy stored in the afternoon and sending electric power to the main grid to obtain greater benefits.

Looking at Figures 7 and 8, we discuss the impact of electricity price on PV-storage charging station operation strategy. When the electricity price and load are low at night, charging stations carry out arbitrage by providing ES charging at a low price and discharging at a high price. From 19:00 to 21:00, the charging demand increases and the PV output is low, and the charging station raises the price, making a profit from the price difference between purchase and sale. Electricity prices are also appropriately raised as demand increases further when PV power generation increases from 21:00 to 00:00. The CSO benefits mainly from low-cost PV power generation and ES. From 12:00 to 18:00, the electricity price fluctuates steadily, controlling the stable fluctuation of the charging load. At this time, the revenue mainly comes from PV power generation and the price difference between purchase and sale. At night, since the main grid has a high electricity purchase price and no PV power generation, PV charging stations raise their electricity prices and reduce some of the load. Charging profits mainly come from the discharge of ES.

Figure 9 shows the daily SOC of the ES system that can reflect its operating state in the PV-storage charging station. SOC can be roughly divided into the following parts. The ES system has rapidly changing SOC at night. As seen by the operating state in Figure 9, it can be concluded that the PV-storage charging station utilizes the nighttime price spread by performing fast charging and discharging to maximize profits. From 03:00 to 05:00, the ES system is in the fast charging state and the energy storage reaches the upper limit of SOC, then is maintained at 56% after proper discharge. From 09:00 to 11:00, the discharge energy storage is in a lower state and is recharged at 11:00 and 13:00 to the upper limit of SOC, then is discharged in the afternoon to maintain at 56%. At night, it is discharged again to remain at the lower limit of SOC.

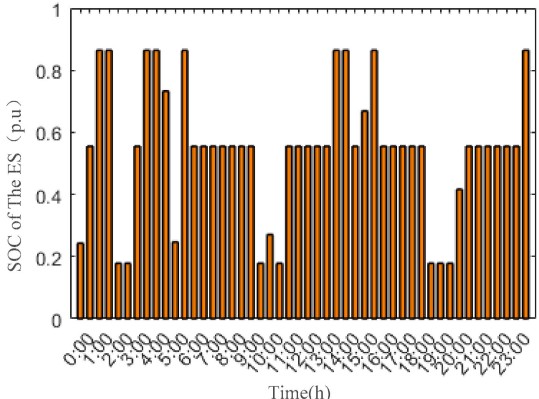

**Figure 9.** SOC of the ES.

Figure 10 shows the PV/load ratio of the PV-storage charging station. From this figure, it can be seen that PV has a contribution rate of over 40% from 10:00 to 17:00, which indicates that the charging station can make full use of PV power generation. The reasonable combination of PV and ES not only improves the charging station's economy but also better consumes and utilizes renewable energy.

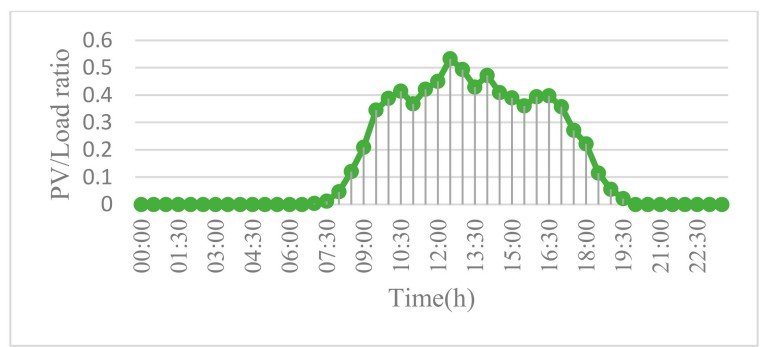

**Figure 10.** PV/load ratio of PV-storage charging station.

The following conclusions can be drawn by analyzing the above results. First, in the case of mature technology and reduced cost, PV-storage charging stations should make full use of space to invest in and construct photovoltaic and energy storage facilities. Secondly, CSOs should set reasonable electricity prices and adjust the charging demand of EV users via electricity pricing means to achieve balanced benefits. Finally, CSOs need to make full use of the characteristics of photovoltaic power generation and especially energy storage, and properly adjust charging and discharging based on the electricity price to promote the economic efficiency of system operation.

*4.3. Models Comparison*

This section compares and analyzes the effects of the bi-level model and the common single-level planning model. The single-level model considers that the charging demand of EV users $d_{v,t}^{ev}$ is fixed and is equal to the calculated demand for the bi-level optimization model, and assumes that the CSO can accurately predict EV user types and the number of electric vehicles at different time points in advance, i.e., every EV user has a fixed charging demand, regardless of choice.

Based on the above assumptions and design, this section constructs two schemes, shown in Table 4. Scheme I builds a single-level model where the CSO's electricity sales price is set at 0.35 \$/kWh. Scheme II is a bi-level model that takes account of capacity configuration and electricity price optimization under uncertainty conditions.

**Table 4.** Scheme settings.

|  | Scheme I | Scheme II |
| --- | --- | --- |
| Planning type | Single-level planning | Bi-level planning |
| Electricity price | 0.35 \$/kWh | Optimized price |

The results and economic characteristics of schemes using CPLEX are shown in Table 5.

**Table 5.** Planning results of different schemes.

|  | Scheme I | Scheme I Without ES | Scheme II |
| --- | --- | --- | --- |
| CSO electricity price (\$/kWh) | 0.35 | 0.35 | Figure 7 |
| CF installed power (kW) | 1154 | 1154 | 862 |
| PV installed power (kW) | 500 | 500 | 500 |
| ES installed power (kW) | 846 | - | 545 |
| ES installed storage capacity (kWh) | 900 | - | 900 |
| Investment cost (\$) | 87,376 | 56,691 | 79,650 |
| Operation and maintenance cost (\$) | 13,644 | 12,924 | 11,839 |
| Total revenue (\$) | 802,991 | 590,572 | 844,029 |
| Net revenue (\$) | 701,971 | 520,975 | 752,485 |
| PV output (kWh) | 2625 | 2625 | 2625 |
| PV output (%) | 22.6% | 22.6% | 29.9% |

As can be seen from Table 5 and Figure 11, different planning schemes result in different configurations of PV-storage charging stations. More specifically, compared with the bi-level model of Scheme II, the single-level model of Scheme I has a larger CF installation capacity and the electricity price is fixed, and the larger installation capacity leads to increased investment operation and maintenance costs and decreased net revenue. The reason for these results is that the charging demand of EV users in the single-level model is set as a fixed value, regardless of electricity price. The CSO needs to invest in and build more charging facilities to meet the needs of EV users, so compared with the EV user choice-based model, this scheme will lead to increased cost.

Compared with Scheme I, in Scheme I without ES, the investment cost is reduced, but the operating return and total revenue are also reduced, which indicates that the installation of ES is beneficial to the charging station. The reason is that the charging station can use ES derived from the power grid or PV and discharge electricity for EV users, obtaining revenue through price differences.

In addition, the PV investment for both schemes is the set upper limit, indicating that the extensive installation of PVs within a limited capacity is beneficial to obtain greater revenue.

The above comparison reveals the importance of EV user behavior in the optimal design of PV-storage charging stations. Since the charging demand of EV users usually varies with the electricity price, it is important to consider user choice when evaluating the profitability of the PV-storage charging station planning scheme.

This section may be divided by subheadings. It should provide a concise and precise description of the experimental results, their interpretation as well as the experimental conclusions that can be drawn.

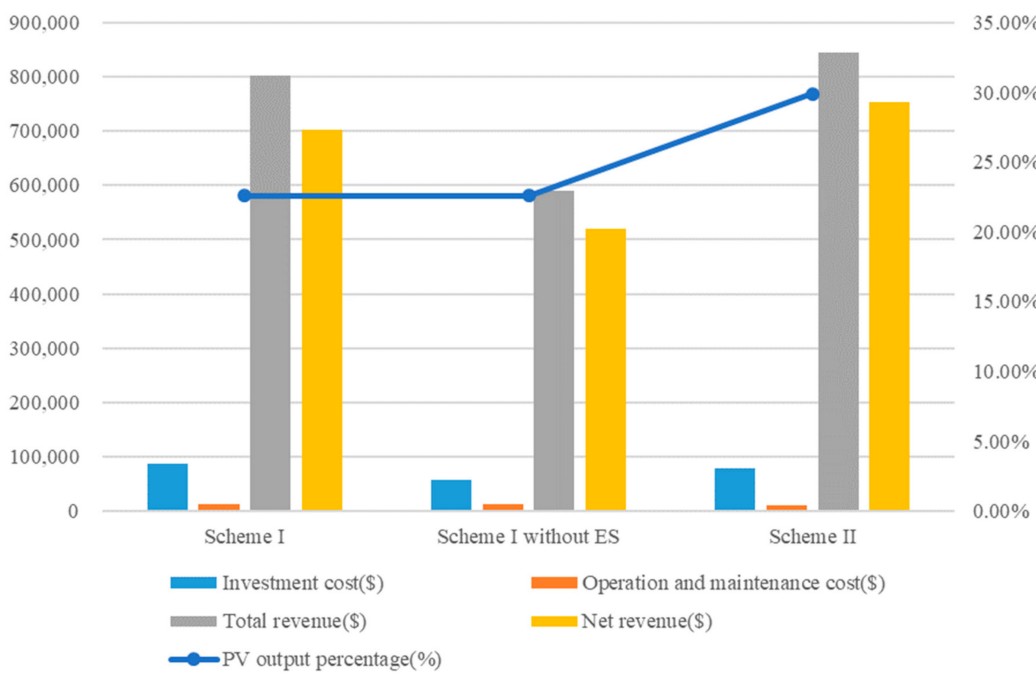

**Figure 11.** Economic effects of the two schemes.

## 5. Conclusions

In order to propose an effective PV-storage charging station planning scheme, this paper established a bi-level planning model for charging stations that considers the needs of EV users. Using dual theory, KKT conditions, and linearization tools, the problem was transformed into mixed-integer linear programming. The model was solved using a commercial solver and a case study was performed. The main research results of this paper are as follows:

(1) We describe and analyze the systematic structure and model framework of a PV-storage charging station. Taking the PV-storage charging station and EV users as the main upper- and lower-level problems, respectively, a bi-level optimization model for capacity allocation and electricity pricing in the PV-storage charging station is established based on constraints such as systematic planning and operation and user demand, with the objective of maximizing revenue and EV user benefits. The bi-level problem is transformed into a single-level mixed-integer linear programming model using the piecewise linear utility function, KKT conditions, and a linearization method.

(2) Using CPLEX (12.7.1, IBM, Armonk, NY, USA, 2018) software for simulation, the validity and practicability of the model are verified by analyzing the planning results, electricity price, operation states, and ES system state.

(3) Through case analysis, it can be concluded that PV-storage charging stations can reduce cost by investing in green PV and carrying out price arbitrage by allocating ES to improve revenue, which is shown in Table 5. By using these methods, the benefits of PV-storage charging stations can be maximized.

This research provides an optimal operational scheme for PV-storage charging stations, including initial investment and construction cost, the output installment capacity of each facility, and a real-time electricity pricing scheme, and has a high economy and practicality.

**Author Contributions:** In this research activity, all authors were involved in the research work, Conceptualization, Y.L. and M.Z.; Methodology, H.D.; Software, H.D.; Validation, Y.L. and S.Z.; Formal Analysis, S.Z.; Investigation, S.W.; Resources, M.L.; Data Curation, M.L.; Writing—Original Draft Preparation, H.D. and S.W.; Writing—Review

& Editing, Y.L.; Visualization, H.D.; Supervision, Y.L.; Project Administration, M.Y.; Funding Acquisition, S.Y. All authors have read and agreed to the published version of the manuscript.

**Funding:** The work described in this paper was supported by National Natural Science Foundation of China (71601078) and the Science and Technology Project of SGCC, Research on the competition situation and operation mode of distribution companies in the park (52170018000S).

**Acknowledgments:** This paper was completed with the help of many teachers and classmates. We would like to express our gratitude to them for their help and guidance.

**Conflicts of Interest:** The authors declare no conflict of interest.

## Nomenclature

Parameters

| | |
|---|---|
| $c^{cf}/c^{rdg}/c^{esp}$ | Unit power investment cost of CF, PV, and ES facilities ($/kW) |
| $c^{cfm}/c^{rdgm}$ | Annual unit operation cost of CF, PV facilities ($/kW/years) |
| $c^{ese}$ | Annual unit operation cost of ES facilities ($/kWh/years) |
| $c^{esm}$ | Annual unit operation cost of ES facilities ($/kWh/years) |
| $d^{ev}_{min}/d^{ev}_{max}$ | Lower/upper limit of EV charge level (kWh) |
| $E^{Nes}_{max}/P^{Nes}_{max}$ | Maximum installed storage capacity/power of ES (kWh; kW) |
| $k^{cf}/k^{rdg}/k^{es}$ | Annualization operators |
| $P^{Ncf}_{max}/P^{Nrdg}_{max}$ | Maximum capacity of installed CF/RDG (kW) |
| $P^{tr}_{max}$ | Rated capacity of distribution transformer (kW) |
| $SOC^{es}_{min}/SOC^{es}_{max}$ | Lower/upper SOC limit for ES (%) |
| $u^{ev}$ | Marginal utility of EV users ($/kWh) |
| $\delta^{ch}_{max}$ | Upper limit of charging tariff ($) |
| $\Delta t$ | Duration of time period (0.5 h) |
| $\eta^{cf}$ | Charging efficiency of CF (%) |
| $\eta^{esc}/\eta^{esd}$ | ES operation efficiency (%) |
| $\theta$ | Number of days in a year |
| $f^{ev}$ | EV uptake |
| $\gamma^{rdg}$ | Ratio of RDG output to installed capacity |
| $\delta^{int}$ | market price ($/kWh) |
| D | Facility lifetime (years) |
| $\zeta$ | Discount rate (%) |

Variables

| | |
|---|---|
| $d^{ev}$ | Required EV charge level (kWh) |
| $E^{es}$ | Stored energy in ES units (kWh) |
| $E^{Nes}/P^{Nes}$ | ES storage capacity/power (kWh; kW) |
| $P^{ch}$ | Total EV charging power (kW) |
| $P^{esc}/P^{esd}$ | ES charging/discharging power (kW) |
| $P^{int}$ | Exchanged power with grid (kW) |
| $P^{Ncf}/P^{Nrdg}$ | Installed capacity of CF/RDG (kW) |
| $P^{rdg}$ | Total power output of RDG (kW) |
| $\delta^{ch}$ | Offered charging tariff ($/kWh) |

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
