# Peer review of "An Optimization Approach Considering User Utility for the PV-Storage Charging Station Planning Process"

_processes, doi:10.3390/pr8010083_

Round 1

Reviewer 1 Report

The authors present a mathematical optimization model for planning and operation of a combined EV charging station that includes PV and battery.

 The authors should mention in the introduction how their scenario of public charging, is reflected in the model, e.g. demand fluctuation, charging time fast charging capability, price.

The problem addressed is complex enough without the planning part, because of the stochastic demand and the fluctuating PV output. The authors should discuss the results in Figure 8, e.g. the impact of the  energy price: for instance in the night the battery is charged from the grid, at 18-20h it is discharged to the grid, meaning the battery is used for arbitrage of energy price mainly.

The figure 8 also shows that all the PV power generated is instantly consumed making the PV-buffering role of the battery negligible. Probably the PV/Load ratio should be changed as part of the dimensioning the charging station.

The model is complex and uses many simplifying assumptions, for instance the EV charging characteristics or pricing (energy instead of service).

A general remark is the need for a notation table where variables and the indices are defined. The equations references are sometimes wrong, e.g. line 192 (14) and (15) should read (15) - (16), in line 197 (17) should be (18), line 201, (18) should be (19) etc. In lines 203-204 it is not clear why charging should be one single period (30 minutes?).

I suggest to discuss the usefulness of the battery if the CSO price is constant (scheme I) and to consider also a system without battery for comparison. As I mentioned before, it seems that the advantage of battery (sef-consumption) becomes higher when more PV power can be buffered in the battery (smaller load or higher PV generation).

Several English errors found throughout the text:

line

97 is excess,

article missing e. 116 (The energy exchange) and many more...

212 benefits are, 222 just need.. do, 247 consumers are, 248 are,

Reviewer 2 Report

Lines 152 to 161 (Chapter 2.2.1) as well as Line 172: The distinction between power, energy, capacity as well as the price (absolute or specific prize per kWh) is not figured out clearly

Abbreviations in Table 1 are not explained

Line 395: Mentioned prize in yuan and not in Dollar

Line 404: The mentioned time period regarding Figure 7 is between 19:00h and 21:00h (not 07:00h – 09:00h)

Line 497-499: Is this a comment?

Equation and table cross references need to be checked again

The unit for energy is written in different ways

Round 2

Reviewer 1 Report

The first review comment have been implemented.

Since the PV generation is at anytime smaller than the load (Fig.10), the battery is not used to buffer PV surplus energy. The added comparison in Table 5, shows that, nevertheless, the battery is used for price arbitrage and improves the net revenue.

The authors may mention this design aspect in their conclusions.
